# Entomo-Virological Surveillance and Genomic Insights into DENV-2 Genotype III Circulation in Rural Esmeraldas, Ecuador

**DOI:** 10.3390/pathogens14060541

**Published:** 2025-05-28

**Authors:** Andrés Carrazco-Montalvo, Diana Gutiérrez-Pallo, Valentina Arévalo, Patricio Ponce, Cristina Rodríguez-Polit, Gabriela Echeverría-Garcés, Josefina Coloma, Victoria Nipaz, Varsovia Cevallos

**Affiliations:** 1Centro de Referencia Nacional de Genómica, Secuenciación y Bioinformática (GENSBIO), Instituto Nacional de Investigación en Salud Pública (INSPI), Quito 170403, Ecuador; acarrazco@inspi.gob.ec (A.C.-M.); dgutierrez@inspi.gob.ec (D.G.-P.); cristinajr84@gmail.com (C.R.-P.); mecheverria@inspi.gob.ec (G.E.-G.); 2Centro de Investigación en Enfermedades Infecciosas y Vectoriales (CIREV), Instituto Nacional de Investigación en Salud Pública (INSPI), Quito 170403, Ecuador; jvarevalo.granda@gmail.com (V.A.); wponce@inspi.gob.ec (P.P.); 3Division of Infectious Diseases and Vaccinology, School of Public Health, University of California, Berkeley, CA 94704, USA; colomaj@berkeley.edu; 4Instituto de Microbiología, Universidad San Francisco de Quito USFQ, Quito 170901, Ecuador; vickyvnipaz@gmail.com

**Keywords:** dengue virus, DENV-2, genotype III Southern Asian-American, *Aedes aegypti*, entomo-virological surveillance, next-generation sequencing, phylogenetic analysis, Ecuador, mosquito-borne viruses, arbovirus genomics

## Abstract

Ecuador, a tropical country with frequent dengue outbreaks, including a surge from 16,017 cases in 2022 to 61,329 in 2024, was the focus of this study. The study was conducted in Borbon, a semi-urban rural town in the Esmeraldas province. Genomic analysis, alongside entomo-virological surveillance, provides valuable insights into DENV-2 genotypes. Five pools of female *Aedes aegypti* mosquitoes from Borbon tested positive for DENV serotype 2 through RT-qPCR. One positive pool (CT = 16.13) was sequenced using Illumina MiSeq, and genotyping was conducted via the Dengue Typing Tool and Maximum Likelihood phylogenetic tree. The genotype assigned was III Southern Asian-American. Comparison with other genomes revealed genetic similarity to a human dengue genome sequenced in 2021, also from Esmeraldas, clustering with genomes reported across the Americas, particularly from Colombia and Venezuela. This study enhances our understanding of dengue virus epidemiology in rural areas, emphasizing the critical role of clinical case surveillance and vector monitoring in guiding evidence-based interventions.

## 1. Introduction

Approximately half of the world’s population is at risk of infection with the dengue virus (DENV) [1]. The *Aedes aegypti* mosquito is the primary vector that transmits the virus. This mosquito-transmitted viral infection is one of the most significant and rapidly spreading diseases worldwide [1]. Over the past 50 years, the number of dengue cases has increased more than 30-fold, and dengue is now considered an endemic disease in about 125 countries [1]. In the Americas, dengue appears in outbreak cycles every 3 to 5 years, typically dominated by one or two serotypes. Notably, in 2019, a substantial peak was observed, with reported cases exceeding 3.1 million. Among these, 28,203 cases were severe, resulting in 1773 deaths [2].

Ecuador has experienced a consistent rise in arbovirus-related infections, particularly DENV [3]. Specific areas within the country have reached hyperendemic viral transmission, and new rural and remote areas have significant outbreaks [4]. Furthermore, severe forms of dengue, previously non-existent, are now becoming more prevalent [3]. Clinical signs of the disease can range from asymptomatic infections to mild or moderate febrile cases. However, there is also, a severe variant of dengue that poses life-threatening danger, characterized by plasma loss and hemorrhagic manifestations [5,6]. In 2022, there were 16,017 confirmed dengue cases in the country. This number increased to 27,838 in 2023, and 61,329 confirmed cases were reported in 2024 [7].

Each of the four dengue serotypes comprises different genotypes classified according to the envelope’s phylogeny [8]. For DENV-1, these include genotypes I, II, III (sylvatic), IV, V, and VI. DENV-2 includes the genotypes Asian-I, Asian-II, Asian/American, American, Cosmopolitan, and sylvatic genotype. DENV-3 includes genotypes I, II, III, IV, and V. Finally, DENV-4 has genotypes I, IIA, IIB, III, and sylvatic genotypes [1,9]. The four main serotypes of DENV share approximately 70% of nucleotide identity, but they are antigenically distinct [10]. Thus, infection with one serotype confers protection mostly to homologous viruses. The different outcomes of dengue fever mostly stem from the interaction of these four serotypes with previous immunity. Cross reactive, partially neutralizing immunity can be detrimental, increasing the risk of severe dengue in secondary heterologous DENV infections [11,12,13].

The transmission of dengue in rural areas is an emerging and complex process, driven by factors such as population expansion, connectivity, and human movement that facilitates the movement of infected individuals. Some evidence indicates a rising seroprevalence in children from rural areas and an increasing incidence of dengue over time [14].

This study aims to report the complete genome sequence, genotyping, and the genomic epidemiology of the DENV-2 virus detected in an *Aedes aegypti* mosquito pool in Esmeraldas, Ecuador.

## 2. Materials and Methods

### 2.1. Vector Study Site

As part of the Asian-American Centers for Arbovirus Research and Surveillance (A2CARES), an active surveillance network and cohort study of arboviruses, our collaborating team (UC Berkeley, CIREV, USFQ) conducts epidemiological surveillance and research in Ecuador. In April 2022, the Center for Research on Vector-borne and Infectious Diseases (CIREV) of the National Institute for Research on Public Health (INSPI) carried out entomological sampling in Borbon, Eloy Alfaro canton, Esmeraldas province, Ecuador (−78.89769; 1.087634). Borbon, a commercial town with approximately 8000 inhabitants, exhibits urban–rural characteristics, including flood-prone areas and low-income neighborhoods (Figure 1).

### 2.2. Sample Collection

Adult *Aedes aegypti* mosquitoes were collected indoors from 122 randomly selected homes in Borbon using a ProkoPack™ aspirator (John W. Hock Company, Gainesville, FL, USA) during morning, noon, and afternoon hours. Mosquitoes were identified on-site using Rueda’s pictorial key (2004) [15]. Female *Aedes aegypti* (fed or unfed) were individually placed in RNAlater^®^ (Thermo Fisher Scientific Inc., Waltham, MA, USA) and grouped into pools containing 10 to 30 individuals each. The samples were transported maintaining the cold chain to the Molecular Biology Laboratory of the CIREV-INSPI located in Quito—Ecuador and stored at −80 °C until processing.

### 2.3. RNA Isolation and RT-qPCR

A total of 155 female *Ae. aegypti* mosquitoes, organized in 8 pools, were processed. Each pool was initially macerated with micro-pistils (Fisher Scientific, Waltham, MA, USA) in liquid nitrogen to extract RNA using the RNeasy Mini Kit (Qiagen, Hilden, Germany), following the manufacturer’s instructions.

The ZDC Multiplex RT-PCR Assay Kit (Bio-Rad, Hercules, CA, USA) was employed to simultaneously detect Zika, Dengue, and Chikungunya viral RNA; with positive controls for each virus. Dengue virus (DENV)-positive pools were further analyzed using the CDC Real Time RT-PCR assay for detection and typing of DENV; with positive controls comprising four DENV serotypes isolated in cell [16].

### 2.4. Sequencing and Genome Assembly

Sequencing and bioinformatics analyses were carried out at the National Reference Center for Genomics, Sequencing, and Bioinformatics (CRN-GENSBIO) at INSPI. The COVIDSeq Assay amplicon (Illumina^®^, San Diego, CA, USA) was employed. Specific primers designed for DENV-2 were used (Appendix A) enabled the PCR amplification of genome fragments ranging of 400 to 600 base pairs (bp). These amplified fragments were subsequently sequenced using the Illumina MiSeq platform (Illumina, San Diego, CA, USA).

Three bioinformatics workflows were used for raw processing .FASTQ reads. These workflows encompassed quality analysis, read trimming, reference mapping (NC_001474.2), and genome assembly. Two of the workflows used similar tools, such as fastQC, Trimmomatic, BWA or Bowtie2, and Ivar consensus. The third workflow followed the ViralFlow protocol [17]. The sequenced sample was uploaded to the GISAID EpiArbo database under accession number EPI_ISL_18195022.

### 2.5. Genotyping and Phylogenetics

Genotyping analysis was conducted using the Dengue Virus Typing Tool. This method utilizes the envelope glycoprotein (E) gene (1.485 bp) and whole genomes corresponding to the following serotypes and genotypes: DENV-1 (genotypes: I, II, III, IV and V), DENV-2 (genotypes: I, American; II, Cosmopolitan; III, Southern Asian-American; IV, Asian II; V, Asian I; and VI, Sylvatic), DENV-3 (genotypes: I, II, III and 3), and DENV-4 (genotypes 4I, 4II, 4III and 4IV). Following serotype assignment, phylogenetic analysis, to determine the phenotype, was performed using the Maximum Likelihood method with the same tool [18]. Additionally, a Multiple Sequence Alignment Viewer (MSA) analysis was performed using BLAST+ 2.16.0 in GenBank to identify similarities with other genomes in this database.

All genomes belonging to the DENV-2 genotype III Southern Asian-American reported for Ecuador, between 2014 and 2023, were selected and downloaded from the GISAID EpiArbo database (https://gisaid.org/). The reference genome (NC_001474.2) was used as an outgroup. The dataset was aligned using MAFFT v7.453 [19] and then refined using AliView v1.28 [20]. Sequences were indexed using Augur (with metadata containing dates and locations), then filtered, aligned, inferred through a Maximum Likelihood tree with 1000 bootstrap replicates, and underwent molecular clock refinement, and the results were exported for display in Auspice [21]. Each resulting tree was formatted in Figtree v1.4.4 [22]. The sequence of DENV-2 isolated from *Ae. aegypti* was analyzed, and a comparative analysis was conducted with sequences from human cases published in GISAID [23].

## 3. Results

### 3.1. DENV Serotyping Isolated from Aedes aegypti Samples

Mosquito samples were collected during the rainy season in April 2022. Five out of eight pools (5/8) tested positive for DENV-2, with CT values between 16.13 and 37.01. We decided to sequence the DENV-2 positive pool with CT value of 16.13; the other four positive pools had CT values greater than 25. The selected pool comprised 30 *Aedes aegypti* mosquitoes (13 unfed and 17 fed) (Appendix A). The selected pool had the lowest CT value (16.13), which indicates high viral load, and was prioritized for sequencing due to limited resources and to ensure optimal genome recovery. This period corresponded with a national increase in dengue cases, aligning with a seasonal outbreak peak.

### 3.2. Genome Assembly and Genotyping

Using different workflows, the entire DENV-2 genome was assembled with 1891 Ns of 10,723 base pairs. The sequenced genome (EPI_ISL_18195022) covered approximately 90% of the reference genome (NC_001474.2) (Figure 2a). The mosquito sample (EPI_ISL_18195022) was assigned to Genotype III—Southern Asian-American (Figure 2b). The MSA Viewer in the BLAST analysis revealed a close relationship with genomes from Venezuela and Colombia posted in GenBank (Appendix A), similar to those reported from human samples in the same area.

### 3.3. Phylogenetics Reconstruction of DENV from Esmeraldas

A phylogenetic analysis was conducted using the reference sequence (NC_001474.2), the sequence obtained from DENV isolated from *Aedes aegypti* (EPI_ISL_18195022), and 73 DENV-2 genotype III Southern Asian-American genomes downloaded from GISAID. which corresponded to human samples from Ecuador. The sequences collected during 2014–2015 are labeled in black (without locality assignment) in the phylogenetic tree (Figure 3) and formed an ancestral clade compared to the rest of the samples. Another clade was formed from the sequences collected during 2020–2023. This recent clade includes samples collected from ten provinces of Ecuador, with 14 samples belonging to the province of Esmeraldas marked in greenish yellow. The genome reported in this study (enclosed in a circle) is closely related to the sample from the Timbire locality sequenced in 2021, with a bootstrap value of 92.3, and forms a genetic group with other samples from nearby localities such as Borbon, Santa Maria, and Colon Eloy, all in the rural Eloy Alfaro canton.

## 4. Discussion

The mosquito collection occurred in April 2022, towards the end of the rainy season and a reported peak in dengue incidence nationwide. Official case reports showed confirmed cases rising over threefold between 2022 and 2024 [7]. The temporal overlap underscores the relevance of our findings and suggests that vector-based surveillance during seasonal peaks can yield early genomic data regarding the circulating serotypes and genotypes. To our knowledge, this is the first full genome sequence of DENV-2 from *Aedes aegypti* directly in rural Esmeraldas during the outbreak of 2022, providing valuable information regarding local viral evolution and transmission patterns.

The present entomo-virological surveillance research demonstrates that it is possible to isolate a complete DENV genome from a maximum of 30 pools of mosquitoes directly. This is consistent with de Figueiredo et al. (2010) [24]. Furthermore, the integrity of the RNA allows for molecular detection and the use of deep sequencing methods. Although a single mosquito pool was sequenced, it was selected on the grounds of a CT value of 16.13—the lowest for the positive pools—representing a high viral RNA load. This approach was taken in order to derive high-quality sequencing output, particularly under financial constraints. Further sequencing can add robustness, but our results are valuable insofar as they give phylogenetic information and also support the validity of entomo-virological genomic surveillance within resource-poor rural settings.

The phylogenetic analysis shows that the DENV-2 isolated from *Aedes aegypti* belongs to genotype III Southern Asian-American. This genotype formed a cluster with genomes from other countries in the Americas, including Belize (FJ898461.1), Nicaragua (GQ199868.1), Guatemala (HQ999999.1), Cuba (AY702036.1), Puerto Rico (EU687217.1, EU687216.1), and the Dominican Republic (AB122020.1), as determined by the Dengue Virus Typing Tool for reference genome assignment. The phylogeographic study published by Allicock et al. (2012) [25], analyzed 191 sequences of this genotype from DENV samples collected in various countries in the Americas between 1981 and 2008. It is estimated that this genotype entered the continent through the Greater Antilles in 1979, subsequently spreading to the Lesser Antilles between 1992 and 1995, and then to the South American coast [26].

During 2019–2021, two events of possible DENV-2 introduction were reported in Ecuador [23]. The first event involved a strain that arrived before 2009 and circulated until 2015. This strain was closely related to Colombian strains that circulated up to 2013. The second event involved a strain that arrived in 2013, resulting in an Ecuadorian lineage that continues to persist [23].

Borbon, located in northwestern Ecuador, is a commercial town situated 117 km from Tumaco (Nariño, Colombia) by land, with connections through rivers and the Borbon–Limones–Tumaco (Colombia) sea route. Therefore, the detection of DENV in mosquitoes and human serum indicates active virus circulation between the neighboring countries. After its arrival to Ecuador, DENV-2 may have spread from Borbon to nearby communities such as Santo Domingo and Santa Maria (localities interconnected by river routes), and from Borbon to Maldonado (17.4 km), Timbire (30.7 km), and Esmeraldas city (107 km) (localities interconnected by land routes). Due to commercial exchange, human migration, and a high mutation rate, the dengue virus undergoes constant evolutionary changes, increasing the likelihood of emergence of new variants. This is particularly relevant in the study area where there is high mobility among populations from Colombia, Ecuador, and more recently from Venezuela [23]. The cluster of our sequenced genome with those from Timbire, Borbon, and surrounding areas supports the hypothesis of localized transmission and the persistence of an established genetic cluster in Eloy Alfaro canton. Despite the high similarity between the genome from the mosquito (EPI_ISL_18195022) and an attested human DENV genome sequenced in 2021 from the same area, no clinical data from human cases accessible throughout the exact period of mosquito collection were accessible for the analysis. Such a shortfall emphasizes the need for future studies to integrate entomo-virological genomic surveillance with clinical case reporting to associate mosquito-derived genotypes with human disease. This amalgamation will enable a more accurate estimation of dengue transmission dynamics within the study area.

In a study by Waman, W. et al. (2016), in silico research showed that the population displaying genotype III Southern Asian-American is divided into four main groups, corresponding to Asia, Central America, South America, and North America [8]. Within this group, seven subpopulations or lineages further reveal subdivision within the South American and Central American groups [8]. These findings are similar to those reported by Fritsch et al. (2023), who also identified DENV-2 genotype III—Southern Asian-American [27]. Furthermore, the geographical dispersion of the DENV-2 serotype may be associated with various mechanisms used to evade or escape the immune response [28].

Phylogenetic reconstruction revealed that sequences from Ecuador collected in 2020, 2021, 2022, and 2023 formed a distinct clade, separate from samples collected in 2014 and 2015, indicating evolutionary events and genetic changes in genotype III Southern Asian-American. Factors such as human population immunity, mosquito vector competence, and capacity, seasonal variations, and random stochastic events contribute to the evolution and genetic changes in DENV [29]. Additionally, this virus acquires one mutation per replication cycle in a vertebrate, with errors attributed to RNA-dependent RNA polymerase (RdRp), resulting in intra host genetic variants [30,31].

In addition to our study, several entomological and virological studies in urban Ecuador have reported striking differences in the dynamics of dengue transmission compared to rural areas. Urban localities such as Guayaquil, Guayas, are more densely populated and have frequent outbreaks due to the presence of more breeding sites for vectors and fewer control measures for vectors [2,7]. For instance, the work of Lee et al. (2021) [4] described the application of unmanned aerial vehicles to identify vector breeding hotspots within cities as a measure of the need for targeted interventions. Similarly, Márquez et al. (2023) [23] reported differences in vector abundance and DENV-2 genetic diversity between urban and rural settings and suggested that surveillance will have to be tailored to each setting’s specific needs. By incorporating such information, we can better understand the dynamics of dengue transmission and enhance the incorporation of entomo-virological surveillance programs to serve both urban and rural areas in order to facilitate more effective preventive measures.

The application of next-generation sequencing (NGS) technology in epidemiological contexts had successfully enabled the complete genome of the dengue virus within a mosquito vector, identifying its genotype. This capability is crucial for understanding the virus’s genetic variability, predicting potential severe outbreaks, and tailoring specific prevention and control strategies specific to the serotypes or genotype present in a particular geographic area. Entomo-virological surveillance plays a critical role in assessing changes in risk and in the effectiveness of control measures. This study enhances the understanding of dengue virus (DENV) epidemiology in rural areas, providing valuable insights for evidence-based decisions in designing efficient surveillance programs and interventions.

In order to further enhance dengue control efficacy in rural areas, we suggest integrating next-generation sequencing (NGS) into national entomo-virological surveillance programs. By sequencing mosquito samples from transmission risk-exposed areas and rural and frontier provinces such as Esmeraldas, the national health authorities will possess early genomic data that can enable early detection of circulating DENV genotypes. This approach would enhance predictive capability for outbreak forecasting and enable targeted vector control interventions. We also recommend strengthening regional collaboration with neighboring countries to improve cross-border mosquito management, the overall regional response to dengue outbreaks.

## Figures and Tables

**Figure 1 pathogens-14-00541-f001:**
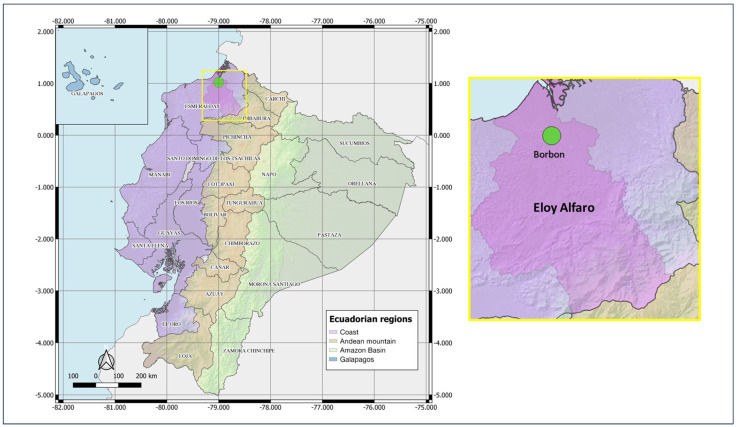
Map of entomological sampling location during 2021–2022. Country: Ecuador; Province: Esmeraldas; Canton: Eloy Alfaro; Locality: Borbon. QGIS 3.10.0.2024 QGIS Geographic Information System. Open Source Geospatial Foundation Project (http://qgis.org).

**Figure 2 pathogens-14-00541-f002:**
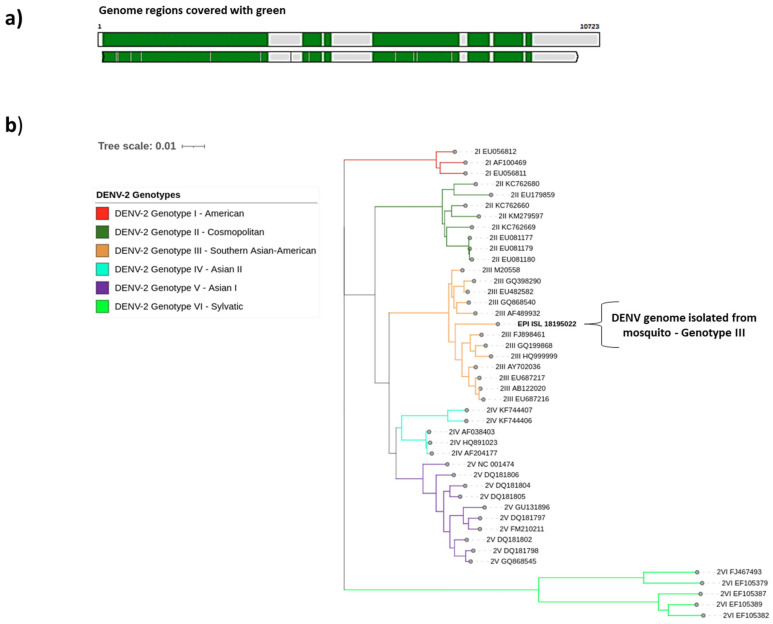
Genome assembly sequenced in this study and genotype assignment. (**a**) Coverage of the sequenced sample EPI_ISL_18195022 against the reference genome NC_001474.2. (**b**) Phylogenetic tree of the DENV-2 genotypes corresponding to the analyzed sample marked with bold (EPI_ISL_18195022). Each genotype is represented by a different color: I—American (red), II—Cosmopolitan (dark green), III—Southern Asian-American (orange), IV—Asian II (light blue), V—Asian I (purple), and VI—Sylvatic (light green).

**Figure 3 pathogens-14-00541-f003:**
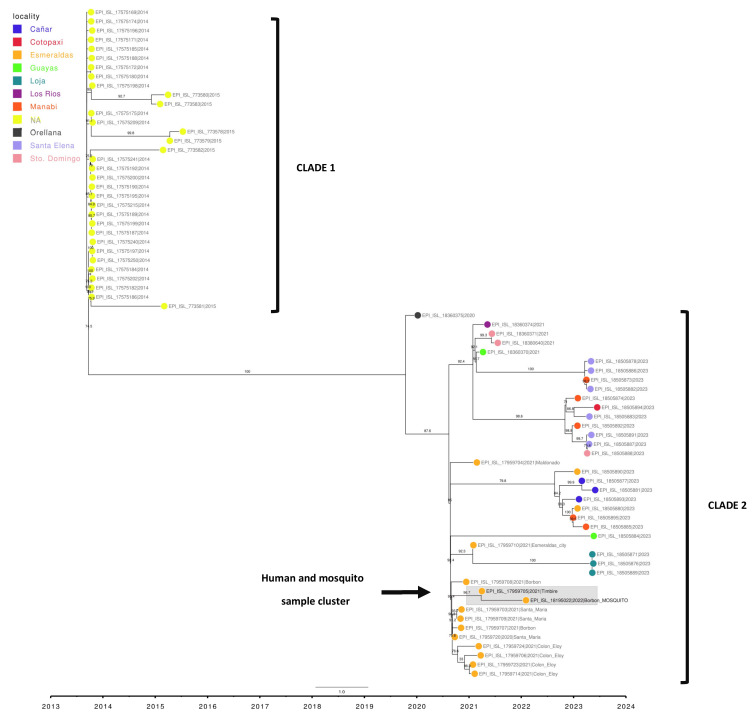
Phylogenetic tree of DENV-2 samples circulating in Ecuador. (Maximum likelihood Tree, Bootstrap: 1000). The sample reported in this study is EPI_ISL_18195022, forming a cluster with EPI_ISL_17959705.

## Data Availability

All sequences analyzed in this study were deposited in GISAID (Global Initiative on Sharing All Influenza Data).

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
