# Peer review of "Entomo-Virological Surveillance and Genomic Insights into DENV-2 Genotype III Circulation in Rural Esmeraldas, Ecuador"

_pathogens, 2025, doi:10.3390/pathogens14060541_

Round 1

Reviewer 1 Report

Comments and Suggestions for Authors

manuscript entitled “Entomo-virological Surveillance and Genomic Insights into DENV-2 Genotype III Circulation in Rural Esmeraldas, Ecuador” by Montalvo et al. offers a valuable contribution to the understanding of dengue virus (DENV) transmission dynamics in Ecuador, combining genomic analysis with entomo-virological surveillance.

I have several comments and suggestions that I believe would strengthen the manuscript:

  1. The abstract currently lacks sufficient context regarding the local dengue situation, demographic structure of the population studied, and the specific analytical tools employed for phylogenetic analysis.
  2. The authors should clarify whether the dengue outbreak coincided with the timing of the entomological and virological surveillance.
  3. The study's sequencing results are based on a single mosquito pool (n = 30), which presents a limitation. It would be helpful for the authors to address why the other pool with a low CT value was not sequenced and discuss the potential implications of this decision on the robustness of the findings.
  4. The authors may consider including entomological and virological studies conducted in Ecuador, particularly focusing on differences observed in urban settings.
  5. The conclusion section could be enhanced by more clearly articulating how the study’s findings might influence dengue control strategies or inform national surveillance efforts. Providing concrete examples or recommendations would significantly improve the applicability of the results.
Comments on the Quality of English Language

Minor English revision is suggested. 

Reviewer 2 Report

Comments and Suggestions for Authors

Improve map resolution (Figure 1)
Only 155 mosquitoes were analyzed in 8 pools, and only 1 was fully sequenced. To assess intra-local diversity, I suggest expanding the sample size and sequencing more positive pools, including those with CT > 25.
The genome found in the mosquito is related to a human genome from 2021, but no parallel clinical data or simultaneously reported cases in the area are presented.
Although the usefulness of genomics is mentioned, no surveillance plans or actions derived from these findings are discussed. Propose how to integrate these data into national entomovirological surveillance strategies or control actions in rural areas.

Reviewer 3 Report

Comments and Suggestions for Authors

This manuscript describes detection and genetic analysis of DENV2 derived from mosquito in Equador. This is interesting study and elaborated, and worth to be shared among scientific and medical communities. Following points should be revised to improve the manuscript.

  1. Line 27: should read "dengue virus epidemiology in rural areas"
  2. Line 74: what is "A2CARES"? Add explanation if it is not mentioned before it.
  3. Line 87: what is "Rueda's pictorial key (2004)"? If necessary, add reference.
  4. Line 120: Add comma, and semicolon.  ex) I, American; II, Cosmopolitan; ...
  5. Figure 2, a) has almost no meaning. Delete, or add any supportive annotation to make it meaningful.
  6. Figure 3 is really difficult to see and understand. Presentation should be revised. Tree is deviated to right side, and strain names can be hardly seen. Is uppermost strain is outgroup? In such case, it can be omitted and shows mostly main branches, with some explanations in legends. The uppermost and lowermost of colors of province is almost same, and readers cannot discriminate. 
  7. Line 206: should read "of emergence of new variants".  
Comments on the Quality of English Language

Minor revisions are necessary.

Round 2

Reviewer 1 Report

Comments and Suggestions for Authors

I appreciate you taking my suggestions into account. At this time, I have no additional feedback. 

Reviewer 2 Report

Comments and Suggestions for Authors

I thank the authors for explicitly stating the limitations and successes of their work, so I give them the go-ahead for publication.